



# A decade of methane measurements at the Boknis Eck Time-series Station in the Eckernförde Bay (Southwestern Baltic Sea)

Xiao Ma[1], Mingshuang Sun[1], Sinikka T. Lennartz[1,2], and Hermann W. Bange[1]

[1] GEOMAR Helmholtz Centre for Ocean Research Kiel, Düsternbrooker Weg 20, 24105 Kiel, Germany
[2] now at ICBM, University of Oldenburg, Oldenburg, Germany

*Correspondence to:* Xiao Ma (mxiao@geomar.de)

**Abstract.** Coastal areas contribute significantly to the emissions of methane ($CH_4$) from the ocean. In order to decipher its temporal variability in the whole water column, dissolved $CH_4$ was measured on a monthly basis at the Boknis Eck Time-series Station (BE) located in the Eckernförde Bay (SW Baltic Sea) from 2006 to 2017. BE has a water depth of about 28 m
and dissolved $CH_4$ was measured at six water depths ranging from 0 to 25 m. In general $CH_4$ concentrations increased with depth, indicating a sedimentary release of $CH_4$. Pronounced enhancement of the $CH_4$ concentrations in the bottom layer (15–25 m) was found during February, May–June and October. $CH_4$ was not correlated with Chlorophyll *a* or $O_2$ over the measurement period. Unusually high $CH_4$ concentrations (of up to 696 nM) were sporadically observed in the upper layer (0–10 m) (e.g. in November 2013 and December 2014) and were coinciding with Major Baltic Inflow (MBI) events. Surface
$CH_4$ concentrations were always supersaturated throughout the monitoring period, indicating that the Eckernförde Bay is an intense but highly variable source of atmospheric $CH_4$. We did not detect significant temporal trends in $CH_4$ concentrations or emissions, despite of ongoing environmental changes such as warming and deoxygenation in the Eckernförde Bay. Overall, the $CH_4$ variability at BE is driven by a complex interplay of various biological and physical processes.

## 1. Introduction

Methane ($CH_4$) is an atmospheric trace gas which contributes significantly to global warming (IPCC, 2013) and the evolution of stratospheric ozone (WMO, 2018). Atmospheric $CH_4$ mole fractions have been increasing by about 150 % since the industrial revolution (IPCC, 2013).

The oceanic release of $CH_4$ to the atmosphere plays a minor role for the global atmospheric $CH_4$ budget (Saunois et al. 2016). However, coastal areas have been identified as hot spots of $CH_4$ emissions (see e.g. Bange et al., 1994; Upstill-Goddard et al.,
2000; Borges et al., 2016). Dissolved $CH_4$ in coastal waters is mainly resulting from the interplay of (i) sedimentary sources such as anaerobic methanogenesis during the decomposition of organic matter (Xiao et al., 2018; Dale et al., 2019) or seepage from oi land natural gas reservoirs (Bernard et al., 1976; Hovland et al., 1993; Judd et al., 2002) and (ii) microbial $CH_4$ consumption which occurs under oxic conditions in the water column and under anoxic conditions in the sediments (Pimenov et al., 2013; Steinle et al., 2017; Egger et al., 2018). Only recently, Weber et al. (2019) estimated the global





oceanic $CH_4$ emissions to range from 6 to 12 Tg yr$^{-1}$, of which about 0.8–3.8 Tg yr$^{-1}$ were attributed to coastal waters. Occasional studies of the $CH_4$ production and consumption pathways in coastal waters and the associated $CH_4$ emissions to the atmosphere have received increasing attention during the last decades (Bange et al., 1994; Reeburg 2007; Naqvi et al., 2010). However, time-series measurements of $CH_4$ which would allow identifying short- and long-term trends in view of the ongoing environmental changes in coastal regions (such as eutrophication, warming and deoxygenation) are still sparse. In

this paper we present the monthly measurements of $CH_4$ from a time-series station in the Eckernförde Bay (Baltic Sea) during 2006–2017.

Due to severe eutrophication, sediments in the Eckernförde Bay receive large amount of organic matter (Smetacek et al., 1987; Oris et al., 1996; Nittrouer et al., 1998) and thus are active sites of $CH_4$ formation (Schmaljohann, 1996; Whiticar, 2002; Treude et al., 2005; Maltby et al., 2018). Seasonal and inter-annual $CH_4$ variations in concentration, saturation and air-

sea flux density were investigated for more than a decade. The aim of this study was to assess the seasonal dynamics of and the environmental controls on $CH_4$ variability in the Eckernförde Bay which is affected by high nutrient concentrations, increasing water temperatures and ongoing loss of dissolved oxygen (Lennartz et al., 2014).

## 2. Study site

The Boknis Eck (BE) time-series station is one of the oldest continuously conducted marine time-series stations in the world.

The first sampling took place in 1957, and has been conducted on a monthly base with only minor interruptions since then (Lennartz et al., 2014). It is situated in the Eckernförde Bay in the southwestern (SW) Baltic Sea, with a depth of approximately 28 m (Fig. 1). The sediments in the Bay are characterized by high organic matter load and sedimentation rate (Orsi et al., 1996; Whiticar, 2002), which is closely associated with the spring and autumn algae blooms (Smetacek, 1985).

The Baltic Sea has only a limited water exchange with the North Sea through the Kattegat, which makes this area very

sensitive to climate change and anthropogenic impacts. As a result of global warming, the increasing trend for the global sea surface (< 75 m) temperatures (SST) was about 0.11 ℃ per decade (IPCC, 2013), while a net SST increase of 1.35 ℃ was observed in the Baltic Sea during 1982–2006, which is one of the most rapid in large marine ecosystems (Belkin, 2009). Lennartz et al. (2014) reported a warming trend of up to 0.2 ℃ per decade at the BE time-series station for the period of 1957–2013. Nutrients in the Baltic Sea have been increasing until 1980s as a result of the intensive agricultural and industrial

activities, and then started to decline due to effective wastewater control (HELCOM, 2018). However, hypoxia and anoxia have been increasing in the Baltic Sea during the past several decades (Conley et al., 2011; Carstensen et al., 2014). Similar trends in nutrients and $O_2$ were also detected at the BE time-series station (Lennartz et al., 2014), indicating that the Eckernförde Bay is representative for the biogeochemical setting of the SW Baltic Sea. In concert with the declining nutrient concentrations, Chlorophyll *a* concentrations at the BE time-series station were declining as well (Lennartz et al., 2014).

Located close to the bottleneck of the water exchange between the North Sea and the Baltic Sea, the BE time-series station is also sensitive to hydrographic fluctuations such as inflows of saline North Sea Water. There is no riverine input to the





Eckernförde Bay, and thus, the saline water inflow from the North Sea plays a dominant role in the hydrographic setting at BE. Because the inflowing North Sea water has a higher salinity compared to Baltic Sea water, a pronounced summer stratification occurs which leads to the development of a pycnocline at about 15 m water depth. The seasonal stratification

occurs usually from mid-March until mid-September. During this period, vertical mixing is restricted and bacterial decomposition of organic material in the deep layer causes pronounced hypoxia and sporadically occurring anoxia during late summer (Lennartz et al., 2014). Pronounced phytoplankton blooms occur regularly in autumn (September–November) and spring (February–March) and to a lesser extent during summer (July–August) (Smetacek et al. 1985).

## 3. Methods

### 3.1 Sample collection and measurement

Monthly sampling of $CH_4$ from the BE time-series station started in June 2006. Seawater was collected from 6 depths (1, 5, 10, 15, 20 and 25 m) with 5 L Niskin bottles mounted on a CTD rosette. 20 mL brown glass vials were filled in triplicates without any bubbles. The vials were sealed immediately with rubber stoppers and aluminum caps. These samples were poisoned with 50 μL saturated aqueous mercury chloride ($HgCl_2$) solution as soon as possible, and then stored in a cool, dark

place until measurement. The storage time of the samples before the measurements was less than 3 months.

A static headspace-equilibrium method was adopted for the $CH_4$ measurements. A 10 mL Helium (99.9999 %, AirLiquide, Düsseldorf, Germany) headspace was created inside the vial with a gas-tight syringe (VICI Precision Sampling, Baton Rouge, LA). The sample was vibrated with Vortex (G-560E, Scientific Industries Inc., New York, USA) for approximately 20 s and then left for at least 2 h to reach the $CH_4$ equilibrium between air and water phases. A 9.5 mL subsample of

headspace was injected into a gas chromatograph equipped with a flame ionization detector (GC-FID, Hewlett-Packard 5890 Series II, Agilent Technologies, Santa Clara, CA, USA). Separation took place on a packed column (SS, 1.8 m length, packed with molsieve 5A, Grace, Columbia, Maryland, USA). Standard gas mixtures with varying mole fractions of $CH_4$ in synthetic air (Deuste-Steininger GmbH, Mühlhausen, Germany and Westfalen AG, Münster, Germany) were used daily to calibrate the response of FID before measurements. The standard gas mixtures were calibrated against NOAA primary gas

standard mixtures in the laboratory of the Max-Planck-Institute for Biogeochemistry in Jena, Germany. Further details about the measurements and calculations of the dissolved $CH_4$ concentration can be found in Bange et al. (2010). The mean precision of the $CH_4$ measurements, calculated as the median of the estimated standard errors (see David, 1951) from all triplicate measurements, was ± 1.3 nM. Samples with an estimated standard error of >10 % were omitted. Dissolved $O_2$ concentrations were measured with Winkler titrations, and Chlorophyll *a* concentrations were measured with a Fluorometer

(Grasshoff et al., 1999). Secchi depth was measured with a white disk (~30 cm in diameter). Sea levels were measured at Kiel-Holtenau, which is about 15 km away from the BE time-series station (http://www.boos.org/).





### 3.2 Calculation of saturation and air-sea flux density

The CH$_4$ saturation (S$_{CH4}$, %) was calculated as:

$$S_{CH4} = 100 \times CH_{4obs}/CH_{4eq} \tag{1}$$

where CH$_{4obs}$ and CH$_{4eq}$ are the observed and equilibrium concentrations of CH$_4$ in seawater, respectively. CH$_{4eq}$ was calculated with the in-situ temperature and salinity of seawater (Wiesenburg and Guinasso, 1979), and the dry mole fraction of atmospheric CH$_4$ at the time of sampling, which was derived from the monthly atmospheric CH$_4$ data measured at Mace Head, Ireland (AGAGE, http://agage.mit.edu/).

The air-sea CH$_4$ flux density (F$_{CH4}$, in µmol m$^{-2}$ d$^{-1}$) was calculated as:

$$F_{CH4} = k \times (CH_{4obs} - CH_{4eq}) \tag{2}$$

where k (in cm h$^{-1}$) is the gas transfer velocity calculated with the equation given by Nightingale et al. (2000), as a function of the wind speed and the Schmidt number (Sc). Sc was computed with the empirical equations for the kinematic viscosity of seawater (Siedler and Peters, 1986) and the diffusion coefficients of CH$_4$ in water (Jähne et al., 1987). Wind speed data were recorded at the Kiel Lighthouse (www.geomar.de/service/wetter/), which is approximately 20 km away from the BE time-105 series station. The wind speeds were normalized to the height of 10 m (u$_{10}$) with the method given by Hsu et al. (1994).

### 4. Results and discussion

### 4.1 Seasonal variations of environmental parameters and dissolved CH$_4$

Seasonal hypoxia were observed every year at the BE time-series station during 2006–2017 (Fig. 2). O$_2$ depletion was detected in the bottom layer (~15–25m) during July–October with minimum O$_2$ concentrations usually occurring in 110 September (Fig. 3). Lennartz et al. (2014) found a significant decrease in dissolved O$_2$ concentrations in the bottom water at the BE time-series station over the past several decades and suggested that temperature-enhanced O$_2$ consumption and a prolonged stratification period might be the causes of deoxygenation. Anoxia with the presence of hydrogen sulfide (H$_2$S) in the period of concurrent CH$_4$ measurements were found in the autumn of 2007, 2014 and 2016, respectively. The anoxic event in 2016 lasted from September until November and was the longest ever recorded at the BE time-series station. In 115 September 2017, a pronounced undersaturation of O$_2$ (~50%) was observed in surface water (Fig. 2). The low temperature together with enhanced salinity in the surface water in September 2017 suggests the occurrence of an upwelling event, which transported O$_2$-depleted and colder bottom waters to the surface. Similar events were also detected in September 2011 and 2012.

Enhanced Chlorophyll *a* concentrations, which can be used to indicate phytoplankton blooms, were usually observed in 120 spring or autumn, but not in every year (Fig. 2). Seasonal variations of Chlorophyll *a* concentrations were generally consistent with the annual plankton succession reported by Smetacek (1985). During 2006–2017, high Chlorophyll *a* concentrations were usually found in the upper layers in March (Fig. 3), which is different from the seasonality during 1960–



2013 where on average, high concentrations occupied the whole water column (Lennartz et al., 2014). Another difference is that no prevailing 'winter dormancy' of biological activity was observed: Chlorophyll *a* concentrations usually remained

high throughout the autumn–spring period. As a proxy of water transparency, Secchi depth was lowest in March indicating a high turbidity, coincident with the Chlorophyll *a* maximum. Chlorophyll *a* concentrations and Secchi depths have been decreasing over the past decades in the Baltic Sea (Sandén and Håkansson, 1996; Fleming-Lehtinen and Laamanen, 2012; Lennartz et al., 2014), but this trend cannot be identified from the median slope at the BE time-series station during 2006–2017.

$CH_4$ concentrations at the BE time-series station showed strong seasonal and inter-annual variability (Fig. 2). During 2006–2017, dissolved $CH_4$ concentrations ranged between 2.9 to 695.6 nM, with an average of 51.2 $\pm$ 84.2 nM. High concentrations were generally observed in the bottom layer (~15–25 m). Enhanced $CH_4$ concentrations were mainly observed during February, May–June and October (Fig. 3). Steinle et al. (2017) measured aerobic $CH_4$ oxidation at the BE time-series station and found that lowest rates occurred in winter, which might be one of the reasons for the enhanced $CH_4$

concentrations in February.

The $CH_4$ accumulation in May and June can be linked to enhanced methanogenesis fueled by organic matter from the spring algae bloom. Capelle et al. (2019) found a positive correlation between mean monthly $CH_4$ concentrations and Chlorophyll *a* concentrations in the upper layers of time-series measurements from Saanich Inlet. Bange et al. (2010) also reported correlations between seasonal $CH_4$ variation and Chlorophyll *a* or Secchi depth, albeit with a time lag of one month, at the

140 BE time-series station during 2006–2008. Although we did not detect such relationships for the extended measurements during 2006–2017, in 2009 and 2016, when no spring algae blooms were detected, $CH_4$ concentrations in following summer months were lower than average (Fig. 2).

Maximum $CH_4$ concentrations were usually observed in October, at the end of the seasonal hypoxia (Fig. 3). Due to the long-lasting anoxic event, strong $CH_4$ accumulations were observed in autumn 2016 (~600 nM), which are the highest in the

145 bottom layer during 2006–2017. Prevailing for several months, depletion of bottom $O_2$ concentrations exerts a strong influence on the underlying sediment. Maltby et al. (2018) detected a shoaling of the sulfate reduction zone in autumn and enhanced methanogenesis in the sediments at the BE time-series station. Reindl and Bolalek (2012) found similar variations in sedimentary $CH_4$ release in the coastal Baltic Sea. In-situ production in the anoxic bottom water might be a potential $CH_4$ source as well (Scranton and Farrington, 1977; Levipan et al., 2007). We, therefore, suggest that the accumulation of $CH_4$ in

the bottom water in October is caused by its release from the sediments and in-situ production in the overlying water column in combination with the pronounced water column stratification during autumn which prevents ventilation of $CH_4$ to the surface layer.

## 4.2 Enhanced $CH_4$ concentrations in the upper water layer

In agreement with Schmale et al. (2010) and Bange et al. (2010), we found that $CH_4$ concentrations generally increase with

155 water depth, indicating a prevailing release of $CH_4$ from the sediments into the water column in the Baltic Sea (see Sect. 4.1).



Nonetheless, unusual high $CH_4$ concentrations in the upper layers were detected sporadically at the BE time-series station during 2006–2017 (Fig. 2). In November 2013 and March 2014, average $CH_4$ concentrations in the upper waters were 187.2 ± 13.9 nM (1–10 m) and 217.8 ± 1.4 nM (5–10 m), which are about 16 and 5 times higher than those found in the bottom layers, respectively (Fig. 4). The most striking event occurred in December 2014, when $CH_4$ concentrations in the upper
layer (1–15 m) were as high as 692.6 ± 3.4 nM (19,890 ± 115 %), whereas dissolved $CH_4$ in the bottom layer (20–25 m) was ~50 nM. The surface $CH_4$ concentration in December 2014 was the highest observed during 2006–2017. In December 2014, a major Baltic inflow (MBI) event occurred, carrying large amounts of saline and oxygenated water from the North Sea into the Baltic Sea (Mohrholz et al., 2015). It is the third strongest event ever recorded, and an unusual outflow period was detected in the Eckernförde Bay: Sea levels declined since mid-November and reached minimum on 10 December, and then
began to increase with the inflow (Fig. 5). The sampling at the BE time-series station took place on 16 December, during the main inflow period. Extreme weather condition (wind speed >15 m s$^{-1}$) were observed several days before the sampling date, and storm-generated waves and currents could have affected the sediment structures in the Eckernförde Bay (Oris et al., 1996).

The significant decrease in sea level alleviated the static pressure on the sediments. Enhanced $CH_4$ release from the
170 sediments, via gas bubbles or exchange from porewater, leads to the accumulation of $CH_4$ in the water column. Similar hydrostatic pressure effects were also reported in tidal systems such as mangrove creeks and estuaries (see e.g. Barnes et al. 2006; Maher et al., 2015; Sturm et al., 2017). Atmospheric pressure also contributes to the overall pressure on the sediments, but it is not recorded at the BE time-series station and thus was omitted. Lohrberg et al. (2020) identified a pronounced $CH_4$ ebullition event in the Eckernförde Bay in the fall of 2014 as a result of the decline in hydrostatic pressures during a weak
storm. The outflow period of the MBI in 2014 lasted for almost a month, and bulk ebullitions and supersaturated water with $CH_4$ could be anticipated. During the inflow period, large amounts of North Sea water flooded into the Eckernförde Bay and presumably pushed the $CH_4$-enriched water to the surface. A negative correlation was found between salinity and $CH_4$ concentration in the water column (Fig. 4a), indicating that vertical $CH_4$ distributions were linked to the mixing of saline water in the bottom and less-saline water in the upper layers. We suggest that $CH_4$ release driven by hydrostatic pressure
fluctuations and the MBI-associated mixing are responsible for the abnormal $CH_4$ profile in December 2014.

The $CH_4$ anomaly in November 2013 can be linked to saline water inflow as well. Nausch et al. (2014) reported the occurrence of an inflow event from 27 October to 7 November in 2013. The sampling at the BE time-series station took place on 5 November, and an increase in salinity was detected in the bottom water (Fig. 4b). The rapid transition from hypoxic to oxic condition in the bottom layer also supports the occurrence of the inflow (Fig. 2). Steinle et al. (2017) found a
185 change in the temperature optimum of aerobic $CH_4$-oxidizing bacteria (MOB) in November 2013 at the BE time-series station and linked it to a displacement of the local MOB community as a result of saltwater injection. Although enhanced $CH_4$ concentrations and high net methanogenesis rates were detected in the sediments in November 2013 (Maltby et al., 2018), the saline inflow with less dissolved $CH_4$ was sandwiched between the sediments and the upper layer waters. As a result, we also found a negative salinity-$CH_4$ correlation in the water column (Fig. 4b). This inflow event was much weaker





than the MBI in December 2014, and no obvious outflow or inflow period can be identified from sea level variations. There was no strong fluctuation in hydrostatic pressure and thus sedimentary $CH_4$ release and $CH_4$ supersaturations in the water column were lower than in December 2014. Another difference is that the decrease in salinity and increase of $CH_4$ concentrations were observed between 10–20 m, which is at shallower depths compared to the MBI in December 2014, indicating that the saline water volume in the bottom layer was larger at the time of the sampling in November 2013.

The situation in March 2014 is different. We did not find any evidence for saline water inflow or hydrostatic pressure fluctuation, and the correlation between $CH_4$ concentration and salinity is poor (Fig. 4c). The occurrences of the unusual $CH_4$ profiles were accompanied by the enhanced Chlorophyll *a* concentrations in the upper waters. $CH_4$ productions by widespread marine phytoplankton have been reported and might be potential sources of surface $CH_4$ supersaturations (Lenhart et al, 2016; Klintzsch et al., 2019). However, spring or autumn algae blooms at the BE time-series station were

often observed without $CH_4$ accumulation and surface $CH_4$ contribution from phytoplankton remains to be proven. Potential sources for the enhanced $CH_4$ in March 2014 are still unclear.

In summary, we suggest that saline water inflow and the subsequent upwelling of water are the most potential causes for the $CH_4$ surface accumulation in November 2013 and December 2014. Nonetheless, the occurrence of inflow does not necessarily lead to enhanced $CH_4$ concentrations in the upper waters. Inflow events are relatively common, for example, in

2013, besides the inflow in November, three other events with similar estimated inflow volumes were detected in January, February and April (Nausch et al., 2014), but no $CH_4$ anomaly was found during that period. The magnitude of the $CH_4$ anomalies might depend on the strength of the inflow events and other factors, such as storms and sediment resuspension. Besides, there is a high chance that the monthly sampling at the BE time-series station only captured few $CH_4$ pulses. Inflow events usually last days to weeks, but the accumulated $CH_4$ in the upper layers might last even shorter because of effective

aerobic $CH_4$ oxidation (Steinle et al., 2017) and strong vertical mixing in winter. The occurrences of surface $CH_4$ accumulations at the BE time-series station might be more frequent than been observed.

### 4.3 Surface saturation and flux density

Surface $CH_4$ saturations are directly proportional to its concentrations in the surface water ($S_{CH4}$=31.40 × [$CH_4$] + 10.29, $R^2$=0.9794, n=77, p<0.0001; Fig. 6a, b), despite of the pronounced seasonal variations in temperature. This indicates that the

215 net $CH_4$ production at BE is overriding the temperature-driven variability of the $CH_4$ concentrations. Excluding the extreme value from December 2014, surface $CH_4$ saturations at the BE time-series station varied between 129–5563 %, with an average of 615 ±688 %. The surface layer was supersaturated with $CH_4$ and thus emitting $CH_4$ to the atmosphere throughout the sampling period.

The coastal Baltic Sea, especially the southwestern part, is a hot spot for $CH_4$ emissions. Area-weighted mean $CH_4$

saturations for the entire Baltic Sea (113 % and 395 % in winter and summer 1992, respectively; Bange et al., 1994) were lower than at the BE time-series station. Schmale et al. (2010) extensively investigated dissolved $CH_4$ distributions in the Baltic Sea, and found that surface $CH_4$ supersaturations were stronger in the shallow western areas.



Sea-to-air $CH_4$ flux densities fluctuated between 0.3–746.3 µmol m$^{-2}$ d$^{-1}$, with an average of 43.8 ± 88.7 µmol m$^{-2}$ d$^{-1}$ (excluding the extreme value in December 2014, Fig. 6c). Comparable results in saturation and flux density were observed at

the pockmark sites in the Eckernförde Bay (Bussmann and Suess, 1998). Although surface $CH_4$ saturations in this study are consistent with the previously published results by Bange et al. (2010) (554 ± 317 %), calculated $CH_4$ flux densities in this study are much higher than in Bange et al. (2010) (6.3–14.7 µmol m$^{-2}$ d$^{-1}$). The discrepancy derives from different flux calculation methods. Bange et al. (2010) adopted the equations by Raymond and Cole (2001) with a lower gas transfer velocity, and they used the median of surface $CH_4$ concentrations for computation, which eliminated the extreme values. Our

results are in good agreement with the ones reported by Bange et al. (2010) if we adopt the same method.

$CH_4$ emissions from coastal waters could be roughly considered as the difference between formation and oxidation of $CH_4$ in the water column and sediments. Although sediments are substantial $CH_4$ sources, most $CH_4$ is consumed before evading to the atmosphere (Martens et al., 1999; Treude et al., 2005; Steinle et al., 2017). Treude et al. (2005) compared the potential and field rates of anaerobic oxidation of methane (AOM) in the sediments of the Eckernförde Bay and suggested that the

AOM-mediating organisms are capable of fast response to changes in $CH_4$ supply. Steinle et al. (2017) reported that 70–95 % of dissolved $CH_4$ were effectively removed in the water column during summer stratification. Apart from MBI-driven uplift of $CH_4$-enriched bottom water to the surface (see below), wind-driven upwelling events can lead to a ventilation of the accumulated $CH_4$ to the atmosphere. For example, Gülzow et al. (2013) observed elevated $CH_4$ concentrations in the Gotland Basin as a result of wind-induced upwelling. The influence of upwelling at the BE time-series station, however, is more

prominent due to the shallow water depth. In September 2012 and 2017, when upwelling occurred (see Sect. 4.1), sea-to-air $CH_4$ flux densities were 65.9 µmol m$^{-2}$ d$^{-1}$ and 132.3 µmol m$^{-2}$ d$^{-1}$, respectively, which were about 50 % and 200 % higher than the mean value.

Enhanced $CH_4$ saturations and associated emissions at the BE time-series station were also strongly promoted by saline inflows (see Sect. 4.2). We found very high surface $CH_4$ saturation and flux density in November 2013 and December 2014

(Fig. 6). In December 2014, surface $CH_4$ saturations were as high as 19,770 % and the calculated flux density reached 3104.5 µmol m$^{-2}$ d$^{-1}$. Inflows of saline waters usually occur in winter, when the well-ventilated water column, relatively low $CH_4$ oxidation rates and high wind speeds are all favorable for high $CH_4$ emissions (Wanninkhof, 2014; Steinle et al., 2017). Assuming that there was no continuous mixing or supply of $CH_4$ to the surface layer, it took about 3.3 days for the accumulated $CH_4$ to come back to equilibrium values under the calculated flux density, during which the annual $CH_4$

emissions from the Eckernförde Bay would increase by approximately 66 % in 2014. This is also in line with our speculation in Sect. 4.2 that the monthly sampling at the BE time-series station might have missed some of the short-lived $CH_4$ pulses.

Moreover, methanogenesis in the sediments of the Eckernförde Bay is sufficient for $CH_4$ bubble formation (Whiticar, 2002). Hydrostatic pressure fluctuations associated with saline water inflow could have triggered $CH_4$ seepage and gas bubble plumes from the seafloor to the atmosphere (Wever et al., 2006; Lohrberg et al., 2020). Gas ebullition sites were usually

found accompanied by pockmark structures (Schneider von Deimling et al., 2011) and Jackson et al. (1998) provided sonar evidences for $CH_4$ ebullition in the Eckernförde Bay. However, recently Lohrberg et al. (2020) reported a widespread $CH_4$





ebullition event in the Eckernförde Bay and found no direct linkage between pockmarks and ebullitions. They estimated the bubble-driven $CH_4$ flux during a weak storm in the fall of 2014 was 1916 µmol m$^{-2}$ d$^{-1}$. These findings point to the fact that ebullition might be an important, but highly variable, additional $CH_4$ efflux to the atmosphere. However, our measurements

did not capture gas bubbles and, thus, the estimate of the overall $CH_4$ emissions resulting from the MBI might be too low. In this case, a time-series monitoring of saline inflows and sea level variations, combined with a continuous observation of $CH_4$ variability, especially in winter, are essential in quantifying $CH_4$ emissions from the Eckernförde Bay.

### 4.4 Comparison with other time-series measurements

Besides this study, time-series measurements of $CH_4$ have also been reported from Saanich Inlet (SI), British Columbia,
Canada (Capelle et al., 2019) and ALOHA station in the North Pacific Subtropical Gyre (Wilson et al., 2017).

Located in a seasonally anoxic fjord, the time-series station in SI has a similar hydrographic setting compared to BE, but a deeper water depth (230 m, Capelle et al., 2019). Surface $CH_4$ saturations at SI fell in the lower end of the range observed here for BE (Fig. 7). Despite the fact that the mean surface saturation in SI was higher, $CH_4$ flux densities were much lower than at BE. Since the air-sea exchange approach of Nightingale et al. (2000) was used in both studies, the discrepancy is
resulting from the higher wind speeds at BE. $CH_4$ saturations from ALOHA were only slightly supersaturated (close to the equilibrium saturation) and the flux densities were consequently low as well, which is resulting from the fact that ALOHA is a deep water (~4800 m) station located in the oligotrophic open ocean where potential strong $CH_4$ sources such as sedimentary release or methanogenesis under low $O_2$ in the water column are negligible (Wilson et al., 2017).

Wilson et al. (2017) analyzed the time-series $CH_4$ data from ALOHA during 2008–2016 and observed a decline in the
surface $CH_4$ concentrations since 2013. They attributed the potential decrease in $CH_4$ production to fluctuations in phosphate concentrations. Capelle et al. (2019) also detected a significant decline of $CH_4$ concentrations in the upper water column over time at SI and proposed a link with the shoaling of the boundary of the hypoxic layer. However, no significant trend was detected in $CH_4$ concentrations or flux densities at the BE time-series station (Fig. 6), despite of the relatively long observation period. The different situations can be explained by the shallow water depth in the Eckernförde Bay, which
makes the $CH_4$ distribution sensitive to the variability of its sedimentary release and events such as MBI and wind-driven upwelling.

### 5. Conclusions

The $CH_4$ measurements at the BE time-series station showed a strong temporal variability and variations with depths. A pronounced enhancement of the $CH_4$ concentrations was usually found in the bottom layer (15–25 m) during February,
May–June and October which indicates that the release from the sediments is the major source of $CH_4$. Organic matter and dissolved $O_2$ are usually considered as the main controlling factors for $CH_4$ production and consumption pathways, but we did not detect correlations of $CH_4$ with Chlorophyll $a$ or $O_2$ during 2006–2017.



Obviously non-biological processes such as local wind-driven-upwelling and the inflow of saline North Sea waters play a significant role for the observed variability of $CH_4$ at BE. However, these phenomena, which occur on relatively short time

scales of day or weeks, were not frequently detected; most probably due to the monthly sampling frequency. The surface layer at BE was always supersaturated with $CH_4$ and therefore, BE was a persistent and strong, but highly variable, source of $CH_4$ to the atmosphere. We did not detect significant temporal trends in $CH_4$ concentrations or emissions, despite of ongoing environmental changes (warming, deoxygenation) in the Eckernförde Bay. Overall, the $CH_4$ variability at BE is driven by a complex interplay of various biological (i.e. methanogenesis, oxidation) and physical (i.e. upwelling, inflow events)

processes. Continuous observations at the BE time-series station, with an emphasis on the period when upwelling and saline inflow usually occur is therefore, of great importance in quantifying $CH_4$ variability and the associated emissions as well as for predicting future $CH_4$ variability in the SW Baltic Sea.

*Data availability.* Data are available from the Boknis Eck Database: https://www.bokniseck.de (Bange and Malien, 2020)
and MEMENTO (the MarinE MethanE and NiTrous Oxide database, https://memento.geomar.de (Kock and Bange, 2015).

*Author contributions.* XM, MS, STL, and HWB designed the study and participated in the fieldwork. $CH_4$ measurements and data processing were done by XM, MS and STL. XM wrote the article with contributions from MS, STL and HWB.

*Competing interests.* The authors declare that they have no conflict of interest.

*Acknowledgements.* The authors thank the captain and crew of the RV *Littorina* and *Polarfuchs* as well as many colleagues and numerous students who helped with the sampling and measurements of the BE time-series through various projects. Special thanks to A. Kock for her help with sampling, measurements and data analysis. The time-series at BE was supported

by DWK Meeresforschung (1957–1975), HELCOM (1979–1995), BMBF (1995–1999), the Institut für Meereskunde (1999–2003), IfM-GEOMAR (2004–2011) and GEOMAR (2012–present). The current $CH_4$ measurements at BE are supported by the EU BONUS INTEGRAL project which receives funding from BONUS (Art 185), funded jointly by the EU, the German Federal Ministry of Education and Research, the Swedish Research Council Formas, the Academy of Finland, the Polish National Centre for Research and Development, and the Estonian Research Council. The Boknis Eck Time-Series Station

(www.bokniseck.de) is run by the Chemical Oceanography Research Unit of GEOMAR, Helmholtz Centre for Ocean Research Kiel. The sea level data used in this study was made available by the EMODnet Physics project (www.emodnet-physics.eu/map), funded by the European Commission Directorate General for Maritime Affairs and Fisheries.

*Financial support.* Xiao Ma is grateful to the financial support provided by the China Scholarship Council (grant no. 201306330056) and the BONUS INTEGRAL project (grant no. 03F0773B).



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



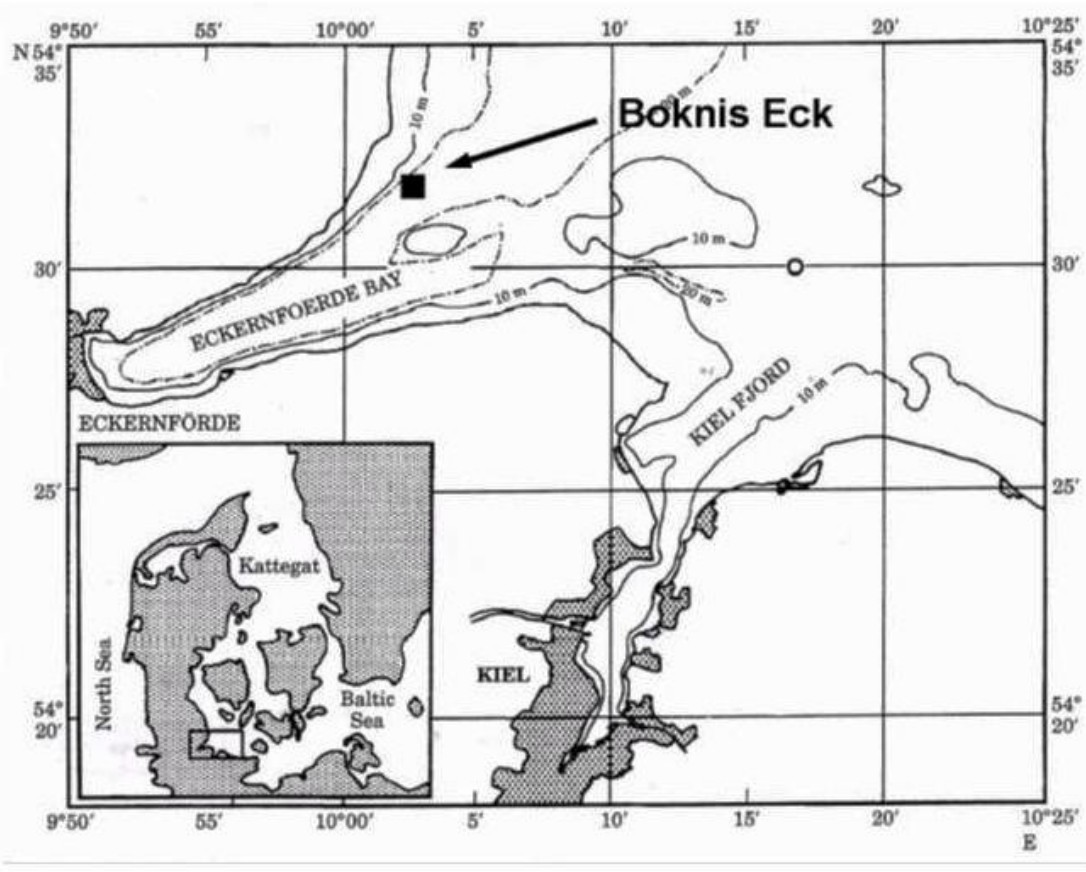

**Fig. 1 Location (black square) of the Boknis Eck time-series station in the Eckernförde Bay, southwestern Baltic Sea. (from Hansen et al., 1999)**





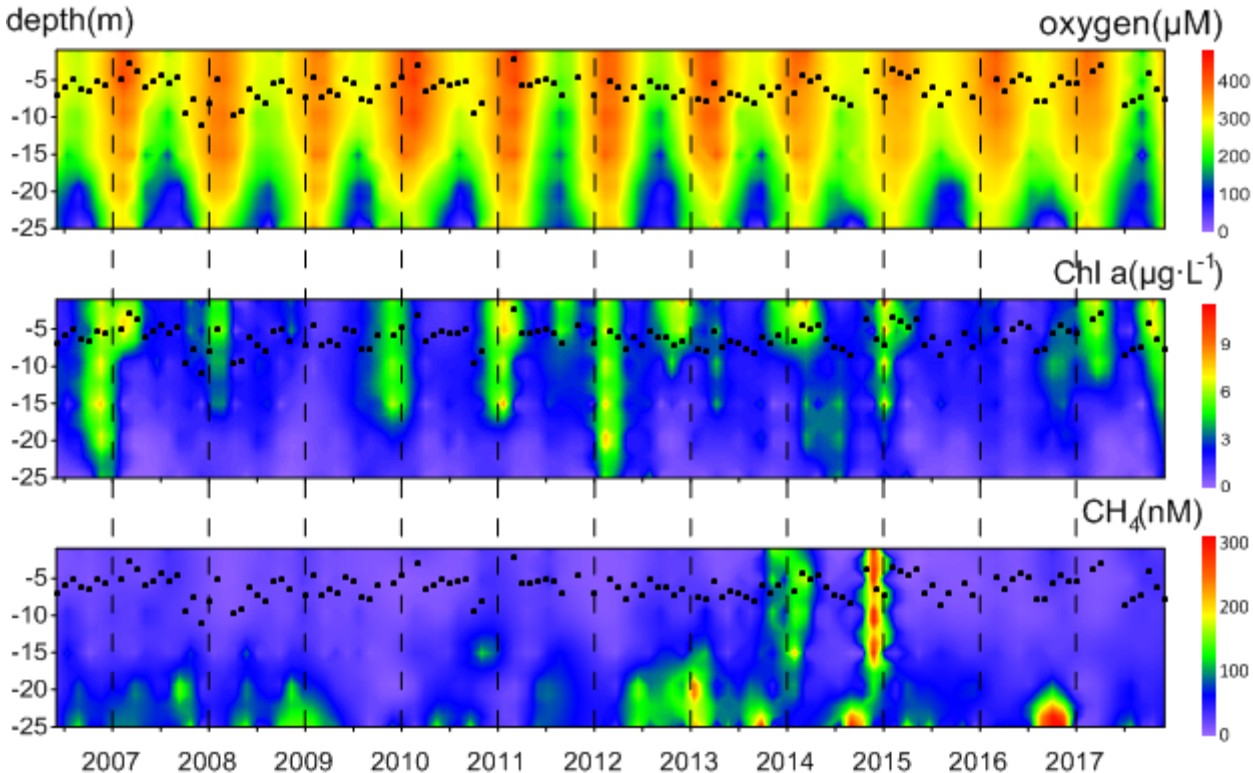

**Fig 2. Distributions of dissolved O$_2$, Chlorophyll *a* and CH$_4$ at the BE time-series station during 2006–2017. Black dots indicate the**
480 **monthly measurements of Secchi depth. To get a better visualization, the maximum color bar for CH$_4$ concentration is 300 nM, but some of the actual concentrations are higher (for example, in December 2014 and in autumn 2016).**





**Fig 3. Mean seasonal variations of dissolved O₂, Chlorophyll *a* and CH₄ at the BE time-series station during 2006–2017. CH₄ concentrations in December 2014 were excluded in plotting.**


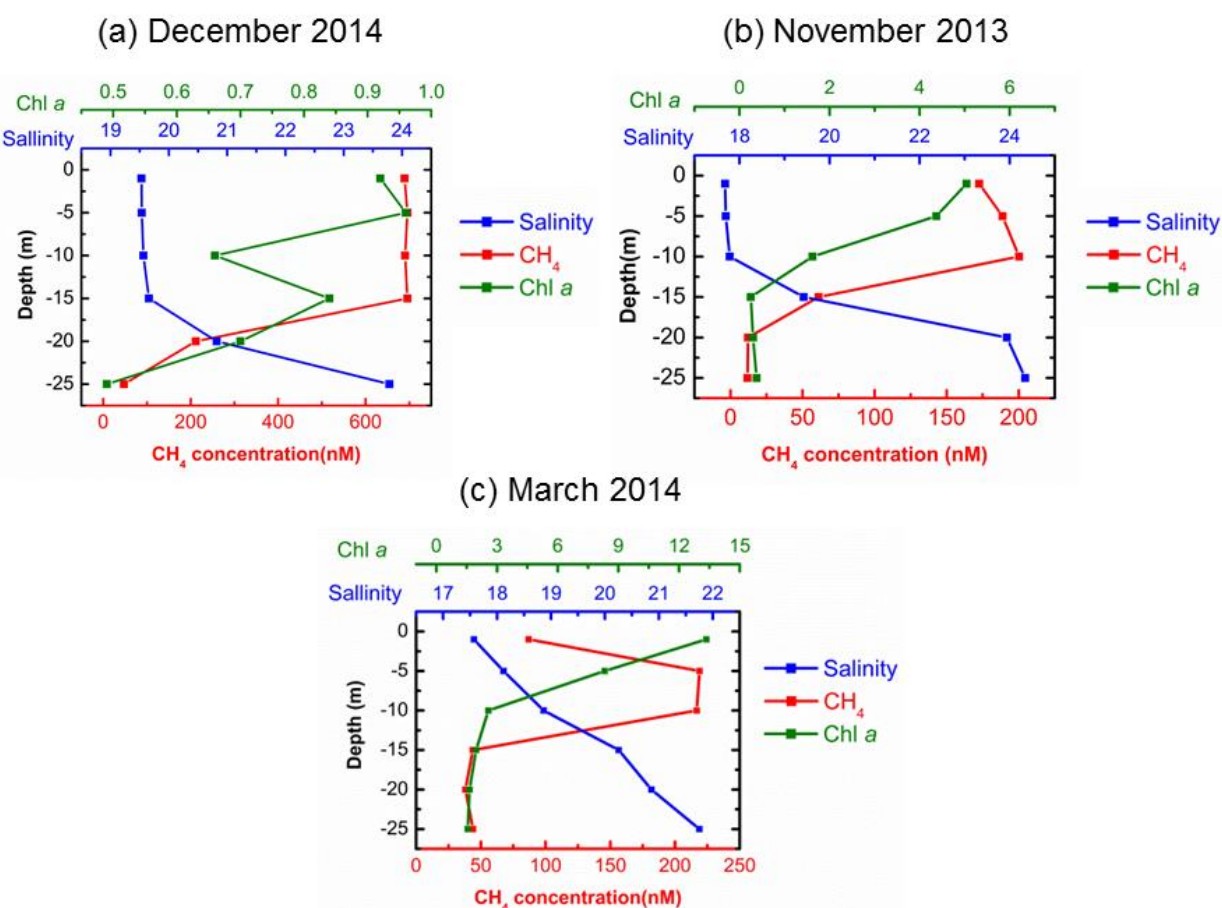

**Fig 4. Vertical distribution of Chlorophyll *a*, salinity and CH₄ concentrations in the water column in December 2014 (a), November 2013 (b) and March 2014 (c).**





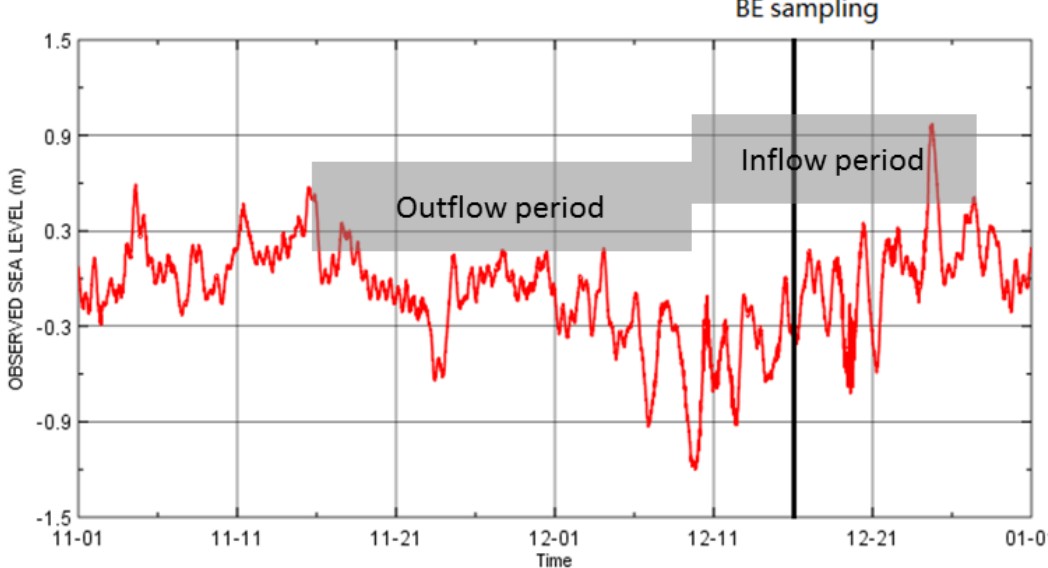

**Fig 5. Sea level variations in November and December, 2014. The black line indicates the occurrence of BE sampling in December 2014.**





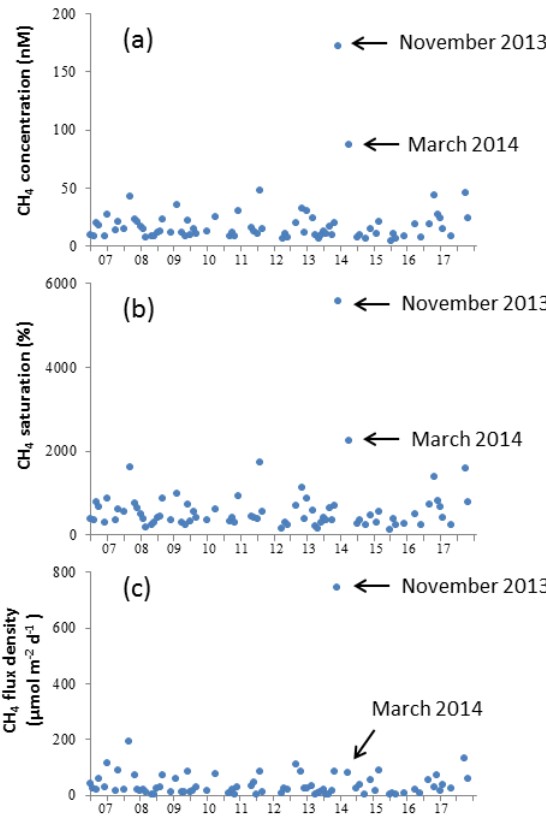

**Fig 6. Inter-annual variations of dissolved CH₄ concentration (a), saturation (b) and flux density (c) at the BE time-series station during 2006–2017. Data collected from December 2014 were not shown.**





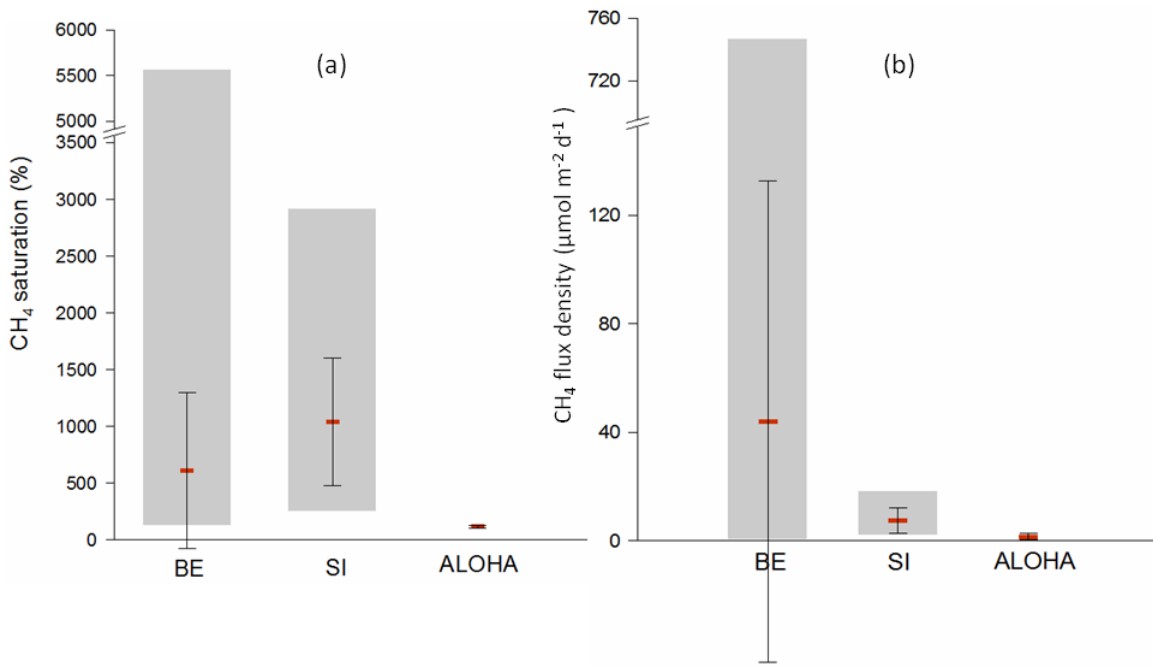

**Fig 7. Comparison of surface CH₄ saturations (a) and flux densities (b) from time-series stations of BE, Saanich Inlet (SI) and ALOHA. For the computation of flux density, the equations of Nightingale et al. (2000) and Wanninkhof (2014) were used for SI and ALOHA, respectively. Data in December 2014 at the BE time-series station were not included. Please note the break on the y**
**axis for both charts.**