# Peer review of "A decade of methane measurements at the Boknis Eck Time-series Station in the Eckernförde Bay (Southwestern Baltic Sea)"

_Biogeosciences, 2020_

## Referee Comment (RC1) · Anonymous Referee #1 · 15 Apr 2020

For characterising marine ecosystem shifts over time, especially in highly anthropogenically impacted regions, sustained time series data are invaluable, but such records are sparse. Their documentation is essential so papers of this type, in this case presenting decadal records of dissolved methane, dissolved oxygen and chlorophyll-a from the Boknis Eck time series site in the Baltic, are welcome. The Boknis Eck site is subject to severe eutrophication and is an active site of methane production so this paper has potential to provide important insights into methane temporal variability. As such this paper clearly falls within the scope of Biogeosciences. The authors represent a group that has a long experience of marine methane measurements and of working at the Boknis Eck site. Their methodology is well established and sound, and

it is described concisely yet in enough detail to enable their reproduction by others. The observations presented are rather straightforward, and while no novel concepts or ideas are described the data are worth reporting and are adequately set into the wider context, citing relevant sources. Overall the paper is well structured and generally easy to follow, and the figures are clear. I was however, a little unclear as to the authors explanation of the unusually high surface methane observed in December 2014. They mention a major inflow at this time, of high salinity, oxygenated North Sea water but it was not clear to me whether they were implying this water to be high or low in methane (or the same) relative to in situ conditions. I think an additional sentence or two would help clarify this. They also describe a major outflow period in which sea levels declined prior to this inflow, and extreme weather that could have affected the sediment structures in the Eckernförde Bay. Presumably this could have led to methane release, but I think they stop short of saying this. Instead, they tend to favour hydrostatic pressure release due to the falling sea level as a cause of methane release from the sediments. It is not especially clear to me how this signal is transferred to the surface. Also, the hydrostatic pressure change, equivalent to the order of 1 metre in a 28-metre water column is rather small relative to the changes that occur in some estuarine and mangrove environments the authors cite. Can they provide evidence that such changes can produce the observations they describe? I wonder how important this mechanism might be relative to other possibilities. It has been documented for example that current flows across the seabed that could be induced by surface inflows in shallow water, can set up pressure gradients driving pore water flow (e.g. Ahmerkamp et al., The impact of bedform migration on benthic oxygen fluxes. JGR Biogeosciences https://doi.org/10.1002/2015JG003106). I think perhaps a little more in-depth discussion of the various possibilities would be insightful. For example, is it possible to estimate the amount of methane that would be expected to be released from the sediments over the duration of the hydrostatic pressure drop, and is this consistent with the observed effect? The authors could perhaps also clarify why they chose to use a different equation for calculating flux densities (Nightingale et al., 2000) to that used

in their earlier paper (Bange et al. (2010), i.e. Raymond and Cole (2001), which gives a lower gas transfer velocity. The authors point out that the two sets of results agree if the same equation is adopted but I was curious about their reasoning in selecting Nightingale et al (2000) for this study. I am not suggesting they are incorrect in this, rather I just wanted to know their reasoning.

———————————————————

---

## Referee Comment (RC2) · Anonymous Referee #2 · 27 Apr 2020

GENERAL COMMENTS

The paper by Ma et al. titled: "A decade of methane measurements at the Boknis Eck Time-series Station in the Eckernförde Bay (Southwestern Baltic Sea)" investigated the CH4 temporal variability (from 2006 to 2017) in the whole water column at the Boknis Eck Time-series Station located in the Eckernförde Bay (SW Baltic Sea). In this system the concentration of CH4 increases with depth due principally to the fluxes from the sediments. Sporadic elevated CH4 concentrations (up to 696 nM) have been observed in the upper layer coinciding with Major Baltic Inflow events. During the period studied the Eckernförde Bay is an intense but highly variable source of atmospheric CH4. The

manuscript is very interesting and as the authors state, time-series measurements of CH4 are still sparse, reason why the study can contribute to have a better knowledge of the behaviour of this greenhouse gas in coastal systems, hot spots of CH4 emissions.

The data are well presented and the discussion of the dataset is comprehensive and conclusive. However, from my point of view, I have some suggestions to render the work more attractive to readers. Therefore, I suggest its publication after minor revisions. Since part of the behaviour of CH4 is attributed to contributions of more saline water from the North Sea and that it is a seasonal study with significant variations in temperature, it is convenient to include the variations of temperature and salinity in Figures 2 and 3. Throughout the manuscript it have been discussing about good and bad correlations between the different variables studied, however, hardly any statistical data (p values, r2) are provided to indicate the good or bad degree of these correlations. I think it would be convenient to include a table with the annual intervals of variation and mean values and deviation of the studied variables including salinity, temperature and wind speed.

SPECIFIC COMMENTS

Pg. 1 Ln 27. Missing "l" in oil. Pg2 Ln 52. Include "temperature increment" in . . ... which is one of the most rapid temperature increment in large marine ecosystems. Pg 3 Ln 74. HgCl2 was added to the sample once it was sealed with rubber stopper and aluminium caps? Was the measurement done with a gas-tight syringe? In that case, could a small pore have been left in the rubber stopper to facilitate gas exchange? Pg 3 Ln 83. The concentrations of CH4 standards used should be indicated, because although the average concentration is $51.2 \pm 84.2$ nM, there were some sporadic samples with very high concentrations (more than 600 nM) and those concentrations should be within the calibration line. Pg 3Ln 89-90. The accurate in dissolved oxygen and Chla measurements should be indicated. How were temperature and salinity measured? What was the precision of these measurements? Pg 4 Ln 112-113. What H2S concentrations were measured? It would be interesting to include these values Pg 4 Ln 115. Indicate the value of the DO concentration that was obtained in the surface waters. This upwelling has also been appreciated in other variables such as nutrients?

Pg 4 Ln 115-116. Since the authors write about behaviour of temperature and salinity, it would be convenient to include graphs of these variables in Figures 2 and 3. Pg 5 Ln 124. What is the reason that in BE the Chla has elevated concentrations only in the upper layers in March and not occupied the whole water column as other works realised in this system? Are Chla and Secchi depth well related to the entire study? Perhaps it could be included in the figure of the Chla the Secchi depth graph. If we look at figure 2 in 2006 and 2012 the Chla occupies the entire water column. What could have happened in these years for the Chla behaviour to be different? Pg 5 Ln 130. To show seasonal and inter-annual variations, a table could also be presented showing the variation interval and annual mean value of each variable. Figures 2 and 3, although very illustrative, have been made with interpolations and do not show the specific data that it is interesting to know. Pg 5 Ln 154. Is there any work in the area where CH4 benthic fluxes have been measured? If so, it would be interesting to include the value. Pg 6 Ln 166-167. Was the water more turbid? Did Secchi's disc reach less depth? Pg 6 Ln 178. What is the r2 of the relation between salinity and CH4? Pg6 Ln 184. Include variation of dissolved oxygen values to change from hypoxic to oxic condition in the bottom layer. Pg 6 Ln 189. Include the correlation coefficient (r2) of the relation between salinity and CH4 in November 2013. Pg 6 Ln 199. Include the correlation coefficient (r2) of the relation between salinity and CH4 in March 2014. Pg 7 Ln 213-214. Since CH4 saturation has been obtained from the surface methane concentration and equilibrium concentrations of CH4 in seawater, it is obvious that surface CH4 saturations are directly proportional to its concentrations in the surface water, I would omit this from the manuscript.

Figures:

In the figures the letters and numbers are in Arial and not in Times New Roman like the rest of the manuscript. Figure 1. The quality of the figure must be improved.

Figures 2 and 3. The axis titles should appear with capital letters as: Depth, not depth and Dissolved oxygen not only oxygen. It should be convenient to include isolines in these figures for a better appreciation of the concentration variations. Figures with temperature and salinity variations should be included.

Please also note the supplement to this comment:
https://www.biogeosciences-discuss.net/bg-2020-107/bg-2020-107-RC2-supplement.pdf

―――――――――――――――――――

---

## Author Comment (AC1) · 1 Jun 2020

We thank reviewer 1 for the helpful comments that helped to improve the manuscript. Please find our replies to the general and specific comments below.

For characterising marine ecosystem shifts over time, especially in highly anthropogenically impacted regions, sustained time series data are invaluable, but such records are sparse. Their documentation is essential so papers of this type, in this case presenting decadal records of dissolved methane, dissolved oxygen and chlorophylla from the Boknis Eck time series site in the Baltic, are welcome. The Boknis Eck site is subject to severe eutrophication and is an active site of methane production so

[Figure]

this paper has potential to provide important insights into methane temporal variability. As such this paper clearly falls within the scope of Biogeosciences. The authors represent a group that has a long experience of marine methane measurements and of working at the Boknis Eck site. Their methodology is well established and sound, and it is described concisely yet in enough detail to enable their reproduction by others. The observations presented are rather straightforward, and while no novel concepts or ideas are described the data are worth reporting and are adequately set into the wider context, citing relevant sources. Overall the paper is well structured and generally easy to follow, and the figures are clear. I was however, a little unclear as to the authors explanation of the unusually high surface methane observed in December 2014. They mention a major inflow at this time, of high salinity, oxygenated North Sea water but it was not clear to me whether they were implying this water to be high or low in methane (or the same) relative to in situ conditions. I think an additional sentence or two would help clarify this.

Reply: Thank you for your suggestion. A direct comparison of the dissolved CH4 concentrations in the North Sea and Baltic Sea would be necessary to assess the impact of the saline water inflow. According to the published results of Bange et al. (1994) and Rehder et al. (1998), CH4 concentrations in surface North Sea is much lower than in the Eckernförde Bay. Advection of water with high CH4 concentration does seem to be unlikely. We thus hypothesize that the MBI led to lower concentrations in the bottom water, substituting previously high concentration throughout the water column in the lower part below the mixed layer, hence causing the observed anomaly in the CH4 concentration profile. We will include the above information and the corresponding references in section 4.2.

They also describe a major outflow period in which sea levels declined prior to this inflow, and extreme weather that could have affected the sediment structures in the Eckernförde Bay. Presumably this could have led to methane release, but I think they stop short of saying this. Instead, they tend to favour hydrostatic pressure release due

to the falling sea level as a cause of methane release from the sediments. It is not especially clear to me how this signal is transferred to the surface.

Reply: We suggest that enhanced CH4 concentrations could be attributed to sedimentary release, and high CH4 concentrations could be either homogeneously distributed all over the water column (via gas bubbles) or only detected at the bottom (via pore-water exchange) when the hydrostatic pressure decreased at first. The CH4-enriched water was subsequently lifted to the surface by the saline inflow, which is heavier than the low salinity-water in the Eckernförde Bay. This is supported by the negative correlation between CH4 concentrations and salinity in the water column. The decline of hydrostatic pressure could be one of the potential causes of the enhanced CH4 release from the sediment. There might be other potential causes, for example, sediment resuspension, resulted either from the storm or the flushing of the strong saline inflow, but this is not supported by the variation of Secchi depths. The occurrence of MBI is usually associated with storms and strong winds, but this is beyond the discussion of this study. We do not have any evidence and therefore, did not discuss the potential impact of the extreme weather conditions. We will add more detail in section 4.2.

Also, the hydrostatic pressure change, equivalent to the order of 1 metre in a 28-metre water column is rather small relative to the changes that occur in some estuarine and mangrove environments the authors cite. Can they provide evidence that such changes can produce the observations they describe? I wonder how important this mechanism might be relative to other possibilities.

Reply: Lohrberg et al. (2020) reported the detection of a widespread CH4 ebullition event in the Eckernförde Bay in October 2014, shortly before the occurrence of the strong MBI. They demonstrated that storm-associated fluctuations of hydrostatic pressure induced the ebullitions and estimated a sedimentary CH4 flux of $\sim$1900 $\mu$mol m-2 d-1, as a result of the changes in water level ($\pm$ 0.5 m) and air pressure ($\pm$ 1500 Pa, equivalent to approximately $\pm$ 0.15 m of water level fluctuation). Air pressure is not recorded at the BE time-series station, and we calculated the sea-to-air flux of $\sim$3100

$\mu$mol m-2 d-1, with the changes in water level of $\pm$ 1 m. Water level fluctuation, when there was no strong wind or inflow event, was approximately $\pm$ 0.2 m in the Eckernförde Bay. Ignoring the CH4 oxidation in the water column, the sharp increase in sea-to-air CH4 fluxes in December 2014 are generally in good agreement with the sedimentary CH4 release reported by Lohrberg et al. (2020), which provides a strong evidence that the changes in water levels are capable of inducing such strong changes in CH4 release. We will incorporate this in section 4.2.

It has been documented for example that current flows across the seabed that could be induced by surface inflows in shallow water, can set up pressure gradients driving pore water flow (e.g. Ahmerkamp et al., The impact of bedform migration on benthic oxygen fluxes. JGR Biogeosciences https://doi.org/10.1002/2015JG003106). I think perhaps a little more in-depth discussion of the various possibilities would be insightful. For example, is it possible to estimate the amount of methane that would be expected to be released from the sediments over the duration of the hydrostatic pressure drop, and is this consistent with the observed effect?

Reply: Thank you for your suggestions. Porewater exchange might be an important benthic CH4 source, and we will add more detail in section 4.2. Sedimentary CH4 release via ebullition from Lohrberg et al. (2020) is generally consistent with our results. Please see the reply above.

The authors could perhaps also clarify why they chose to use a different equation for calculating flux densities (Nightingale et al., 2000) to that used in their earlier paper (Bange et al. (2010), i.e. Raymond and Cole (2001), which gives a lower gas transfer velocity. The authors point out that the two sets of results agree if the same equation is adopted but I was curious about their reasoning in selecting Nightingale et al (2000) for this study. I am not suggesting they are incorrect in this, rather I just wanted to know their reasoning.

Reply: We choose Nightingale et al. (2000) over Raymond and Cole (2001) because

we would like to compare our results with other time-series analysis in section 4.4. As we discussed in section 4.3, there might be a great difference in flux densities originated from the different equations adopted. SI and ALOHA used Nightingale et al. (2000) and Wanninkhof (2014), respectively. Generally fluxes calculated from these 2 equations are close, and we choose the first one because it lies in the middle of many different gas transfer parameterizations, which makes it widely used and well-accepted.

---

## Author Comment (AC2) · 1 Jun 2020

We thank reviewer 2 for the detailed comments. Please find our replies below.

GENERAL COMMENTS The paper by Ma et al. titled: "A decade of methane measurements at the Boknis Eck Time-series Station in the Eckernförde Bay (Southwestern Baltic Sea)" investigated the CH4 temporal variability (from 2006 to 2017) in the whole water column at the Boknis Eck Time-series Station located in the Eckernförde Bay (SW Baltic Sea). In this system the concentration of CH4 increases with depth due principally to the fluxes from the sediments. Sporadic elevated CH4 concentrations (up to 696 nM) have been observed in the upper layer coinciding with Major Baltic
Inflow events. During the period studied the Eckernförde Bay is an intense but highly variable source of atmospheric CH4. The manuscript is very interesting and as the authors state, time-series measurements of CH4 are still sparse, reason why the study can contribute to have a better knowledge of the behaviour of this greenhouse gas in coastal systems, hot spots of CH4 emissions. The data are well presented and the discussion of the dataset is comprehensive and conclusive. However, from my point of view, I have some suggestions to render the work more attractive to readers. Therefore, I suggest its publication after minor revisions. Since part of the behaviour of CH4 is attributed to contributions of more saline water from the North Sea and that it is a seasonal study with significant variations in temperature, it is convenient to include the variations of temperature and salinity in Figures 2 and 3.

Reply: Thank you for your suggestion. Seasonal and inter-annual variations of temperature and salinity will be shown in the figures.

Throughout the manuscript it have been discussing about good and bad correlations between the different variables studied, however, hardly any statistical data (p values, r2) are provided to indicate the good or bad degree of these correlations. I think it would be convenient to include a table with the annual intervals of variation and mean values and deviation of the studied variables including salinity, temperature and wind speed.

Reply: We will include a table with the variables as suggested.

SPECIFIC COMMENTS Pg. 1 Ln 27. Missing "l" in oil. Pg2 Ln 52. Include "temperature increment" in : : :.. which is one of the most rapid temperature increment in large marine ecosystems.

Reply: We will revise them as suggested.

Pg 3 Ln 74. HgCl2 was added to the sample once it was sealed with rubber stopper and aluminium caps? Was the measurement done with a gas-tight syringe? In that case,
could a small pore have been left in the rubber stopper to facilitate gas exchange?

Reply: There is a small pore left in the rubber stopper after poisoning, but the gas exchange is negligible for this standard method. We have tested that the vials are gas-tight despite of the pores.

Pg 3 Ln 83. The concentrations of CH4 standards used should be indicated, because although the average concentration is 51.2 _ 84.2nM, there were some sporadic samples with very high concentrations (more than 600 nM) and those concentrations should be within the calibration line.

Reply: The measurements last for more than a decade and the standard gases we used changed several times. We have adjusted the concentrations of standard gases for every measurement to make sure that the values of the samples fall in the range of the calibration curves. In this case, we think it is not necessary to list all the CH4 standards. We will include this information in the method section.

Pg 3Ln 89-90. The accurate in dissolved oxygen and Chla measurements should be indicated. How were temperature and salinity measured? What was the precision of these measurements?

Reply: Temperature and salinity were measured by the sensors equipped on CTD. Different methods might be adopted for individual parameter during the past decades. A more comprehensive overview of temperature, salinity, dissolved oxygen, Chla as well as other parameters at BE can be found in the paper by Lennartz et al. (Lennartz, S. T., Lehmann, A., Herrford, J., Malien, F., Hansen, H. P., Biester, H., and Bange, H. W.: Long-term trends at the Boknis Eck time-series station (Baltic Sea), 1957–2013: does climate change counteract the decline in eutrophication? Biogeosciences, 11, 6323–6339, https://doi.org/10.5194/bg-11-6323-2014, 2014). We will include this information in the method section.

Pg 4 Ln 112-113. What H2S concentrations were measured? It would be interesting

to include these values

Reply: Unfortunately we did not measure the H2S concentrations. The presence of H2S was recognized by the strong smells of the bottom water. We will add it to the text.

Pg 4 Ln 115. Indicate the value of the DO concentration that was obtained in the surface waters. This upwelling has also been appreciated in other variables such as nutrients?

Reply: We have shown the approximate O2 saturation in the surface water, which is a better proof than the actual DO concentration. The occurrence of the upwelling can be identified by nutrients, too, but not as clearly as indicated by temperature and O2. In this case, we think it is not necessary to include the variation of nutrients in the water column. We will add more detail here.

Pg 4 Ln 115-116. Since the authors write about behaviour of temperature and salinity, it would be convenient to include graphs of these variables in Figures 2 and 3.

Reply: The variations of temperature and salinity will be shown in figure 2 and 3.

Pg 5 Ln 124. What is the reason that in BE the Chla has elevated concentrations only in the upper layers in March and not occupied the whole water column as other works realised in this system? Are Chla and Secchi depth well related to the entire study? Perhaps it could be included in the figure of the Chla the Secchi depth graph. If we look at figure 2 in 2006 and 2012 the Chla occupies the entire water column. What could have happened in these years for the Chla behaviour to be different?

Reply: It remains unclear why elevated Chla concentrations were only detected in the upper layers in March. The overall correlation between Chla and Secchi depth is poor (r2=0.17, p<0.0001, n=111). High Chla concentrations all over the water column in November/December 2006 and March 2012 were coinciding with slightly enhanced nutrients and high temperatures. Nutrients and temperature might be potential environmental controls on Chla distribution. We will incorporate this additional information

in section 4.1.

Pg 5 Ln 130. To show seasonal and inter-annual variations, a table could also be presented showing the variation interval and annual mean value of each variable. Figures 2 and 3, although very illustrative, have been made with interpolations and do not show the specific data that it is interesting to know.

Reply: A table will be shown in the main text as suggested.

Pg 5 Ln 154. Is there any work in the area where CH4 benthic fluxes have been measured? If so, it would be interesting to include the value.

Reply: There are several papers reporting benthic CH4 fluxes in the Eckernförde Bay. Sedimentary CH4 release, via pockmarks or ebullition, was discussed in section 4.3.

Pg 6 Ln 166-167. Was the water more turbid? Did Secchi's disc reach less depth?

Reply: We did not see a strong decline in Secchi depth. We will add this information in the text.

Pg 6 Ln 178. What is the r2 of the relation between salinity and CH4?

Reply: r2=0.84, and this value together with p and n are now included in the text.

Pg6 Ln 184. Include variation of dissolved oxygen values to change from hypoxic to oxic condition in the bottom layer.

Reply: The values are now included as suggested.

Pg 6 Ln 189. Include the correlation coefficient (r2) of the relation between salinity and CH4 in November 2013. Pg 6 Ln 199. Include the correlation coefficient (r2) of the relation between salinity and CH4 in March 2014.

Reply: r2, p and n are now described in the text.

Pg 7 Ln 213-214. Since CH4 saturation has been obtained from the surface methane concentration and equilibrium concentrations of CH4 in seawater, it is obvious that

surface CH4 saturations are directly proportional to its concentrations in the surface water, I would omit this from the manuscript.

Reply: The calculation of CH4 saturation involves concentration, as well as temperature and salinity. The authors would like to point out that surface CH4 saturations are directly proportional to its concentrations, despite of the significant seasonality in temperature and salinity. In another paper about dissolved N2O at BE, the influence of temperature is stronger. We think this comparison might be interesting.

Figures: In the figures the letters and numbers are in Arial and not in Times New Roman like the rest of the manuscript.

Reply: BG does not require a consistency for the fonts in the figures and in the text. As long as the fonts in all of the figures are consistent, it would not be a problem.

Figure 1. The quality of the figure must be improved. Figures 2 and 3. The axis titles should appear with capital letters as: Depth, not depth and Dissolved oxygen not only oxygen. It should be convenient to include isolines in these figures for a better appreciation of the concentration variations. Figures with temperature and salinity variations should be included.

Reply: The figures will be revised as suggested, except for the isoclines. We tried to include isoclines in figure 2 and 3, but they seem crowded and messy because of the strong gradients.

---

## Author Response (AR1)

**Reply to reviewer #1**

We thank reviewer 1 for the helpful comments that helped to improve the manuscript. Please find our replies to the general and specific comments below.

*For characterising marine ecosystem shifts over time, especially in highly anthropogenically impacted regions, sustained time series data are invaluable, but such records are sparse. Their documentation is essential so papers of this type, in this case presenting decadal records of dissolved methane, dissolved oxygen and chlorophylla from the Boknis Eck time series site in the Baltic, are welcome. The Boknis Eck site is subject to severe eutrophication and is an active site of methane production so this paper has potential to provide important insights into methane temporal variability.*

*As such this paper clearly falls within the scope of Biogeosciences. The authors represent a group that has a long experience of marine methane measurements and of working at the Boknis Eck site. Their methodology is well established and sound, and it is described concisely yet in enough detail to enable their reproduction by others.*

*The observations presented are rather straightforward, and while no novel concepts or ideas are described the data are worth reporting and are adequately set into the wider context, citing relevant sources. Overall the paper is well structured and generally easy to follow, and the figures are clear. I was however, a little unclear as to the authors explanation of the unusually high surface methane observed in December 2014. They mention a major inflow at this time, of high salinity, oxygenated North Sea water but it was not clear to me whether they were implying this water to be high or low in methane (or the same) relative to in situ conditions. I think an additional sentence or two would help clarify this.*

Thank you for your suggestion. A direct comparison of the dissolved $CH_4$ concentrations in the North Sea and Baltic Sea would be necessary to assess the impact of the saline water inflow. According to the published results of Bange et al. (1994) and Rehder et al. (1998), $CH_4$ concentrations in surface North Sea is much lower than in the Eckernförde Bay. Advection of water with high $CH_4$ concentration does seem to be unlikely. We thus hypothesize that the MBI led to lower concentrations in the bottom water, substituting previously high concentration throughout the water column in the lower part below the mixed layer, hence causing the observed anomaly in the $CH_4$ concentration profile.

We will include the above information and the corresponding references in section 4.2.

*They also describe a major outflow period in which sea levels declined prior to this inflow, and extreme weather that could have affected the sediment structures in the Eckernförde Bay. Presumably this could have led to methane release, but I think they stop short of saying this. Instead, they tend to favour hydrostatic pressure release due to the falling sea level as a cause of methane release from the sediments. It is not especially clear to me how this signal is transferred to the surface.*

We suggest that enhanced $CH_4$ concentrations could be attributed to sedimentary release, and high $CH_4$ concentrations could be either homogeneously distributed all over the water column (via gas bubbles) or only detected at the bottom (via porewater exchange) when the hydrostatic pressure decreased at first. The $CH_4$-enriched water was subsequently lifted to the surface by the saline inflow, which is heavier than the low salinity-water in the Eckernförde Bay. This is supported by the negative correlation between $CH_4$ concentrations and salinity in the water column.

The decline of hydrostatic pressure could be one of the potential causes of the enhanced $CH_4$ release from the sediment. There might be other potential causes, for example, sediment resuspension, resulted either from the storm or the flushing of the strong saline inflow, but this is not supported by the variation of Secchi depths. The occurrence of MBI is usually associated with storms and strong winds, but this is beyond the discussion of this study. We do not have any evidence and therefore, did not discuss the potential impact of the extreme weather conditions.

We will add more detail in section 4.2.

*Also, the hydrostatic pressure change, equivalent to the order of 1 metre in a 28-metre water column is rather small relative to the changes that occur in some estuarine and mangrove environments the authors cite. Can they provide evidence that such changes can produce the observations they describe? I wonder how important this mechanism might be relative to other possibilities.*

Lohrberg et al. (2020) reported the detection of a widespread $CH_4$ ebullition event in the Eckernförde Bay in October 2014, shortly before the occurrence of the strong MBI. They demonstrated that storm-associated fluctuations of hydrostatic pressure induced the ebullitions and estimated a sedimentary $CH_4$ flux of ~1900 $\mu$mol m$^{-2}$ d$^{-1}$, as a result of the changes in water level ($\pm$ 0.5 m) and air pressure ($\pm$ 1500 Pa, equivalent to approximately $\pm$ 0.15 m of water level fluctuation). Air pressure is not recorded at the BE time-series station, and we calculated the sea-to-air flux of ~3100 $\mu$mol m$^{-2}$ d$^{-1}$, with the changes in water level of $\pm$ 1 m. Water level fluctuation, when there was no strong wind or inflow event, was approximately $\pm$ 0.2 m in the

Eckernförde Bay. Ignoring the $CH_4$ oxidation in the water column, the sharp increase in sea-to-air $CH_4$ fluxes in December 2014 are generally in good agreement with the sedimentary $CH_4$ release reported by Lohrberg et al. (2020), which provides a strong evidence that the changes in water levels are capable of inducing such strong changes in $CH_4$ release.

We will incorporate this in section 4.2.

*It has been documented for example that current flows across the seabed that could be induced by surface inflows in shallow water, can set up pressure gradients driving pore water flow (e.g. Ahmerkamp et al., The impact of bedform migration on benthic oxygen fluxes. JGR Biogeosciences https://doi.org/10.1002/2015JG003106). I think perhaps a little more in-depth discussion of the various possibilities would be insightful. For example, is it possible to estimate the amount of methane that would be expected to be released from the sediments over the duration of the hydrostatic pressure drop, and is this consistent with the observed effect?*

Thank you for your suggestions. Porewater exchange might be an important benthic $CH_4$ source, and we will add more detail in section 4.2. Sedimentary $CH_4$ release via ebullition from Lohrberg et al. (2020) is generally consistent with our results. Please see the reply above.

*The authors could perhaps also clarify why they chose to use a different equation for calculating flux densities (Nightingale et al., 2000) to that used in their earlier paper (Bange et al. (2010), i.e. Raymond and Cole (2001), which gives a lower gas transfer velocity. The authors point out that the two sets of results agree if the same equation is adopted but I was curious about their reasoning in selecting Nightingale et al (2000) for this study. I am not suggesting they are incorrect in this, rather I just wanted to know their reasoning.*

We choose Nightingale et al. (2000) over Raymond and Cole (2001) because we would like to compare our results with other time-series analysis in section 4.4. As we discussed in section 4.3, there might be a great difference in flux densities originated from the different equations adopted. SI and ALOHA used Nightingale et al. (2000) and Wanninkhof (2014), respectively. Generally fluxes calculated from these 2 equations are close, and we choose the first one because it lies in the middle of many different gas transfer parameterizations, which makes it widely used and well-accepted.

**Reply to reviewer #2**

We thank reviewer 2 for the detailed comments. Please find our replies below.

*GENERAL COMMENTS*

*The paper by Ma et al. titled: "A decade of methane measurements at the Boknis Eck Time-series Station in the Eckernförde Bay (Southwestern Baltic Sea)" investigated the CH4 temporal variability (from 2006 to 2017) in the whole water column at the Boknis Eck Time-series Station located in the Eckernförde Bay (SW Baltic Sea). In this system the concentration of CH4 increases with depth due principally to the fluxes from the sediments. Sporadic elevated CH4 concentrations (up to 696 nM) have been observed in the upper layer coinciding with Major Baltic Inflow events. During the period studied the Eckernförde Bay is an intense but highly variable source of atmospheric CH4. The manuscript is very interesting and as the authors state, time-series measurements of CH4 are still sparse, reason why the study can contribute to have a better knowledge of the behaviour of this greenhouse gas in coastal systems, hot spots of CH4 emissions.*

*The data are well presented and the discussion of the dataset is comprehensive and conclusive. However, from my point of view, I have some suggestions to render the work more attractive to readers. Therefore, I suggest its publication after minor revisions.*

*Since part of the behaviour of CH4 is attributed to contributions of more saline water from the North Sea and that it is a seasonal study with significant variations in temperature, it is convenient to include the variations of temperature and salinity in Figures 2 and 3.*

Thank you for your suggestion. Seasonal and inter-annual variations of temperature and salinity will be shown in the figures.

*Throughout the manuscript it have been discussing about good and bad correlations between the different variables studied, however, hardly any statistical data (p values, r2) are provided to indicate the good or bad degree of these correlations. I think it would be convenient to include a table with the annual intervals of variation and mean values and deviation of the studied variables including salinity, temperature and wind speed.*

We will include a table with the variables as suggested.

*SPECIFIC COMMENTS*

*Pg. 1 Ln 27. Missing "l" in oil. Pg2 Ln 52. Include "temperature increment" in : : :.. which is one of the most rapid temperature increment in large marine ecosystems.*

We will revise them as suggested.

*Pg 3 Ln 74. HgCl2 was added to the sample once it was sealed with rubber stopper and aluminium caps? Was the measurement done with a gas-tight syringe? In that case, could a small pore have been left in the rubber stopper to facilitate gas exchange?*

There is a small pore left in the rubber stopper after poisoning, but the gas exchange is negligible for this standard method. We have tested that the vials are gas-tight despite of the pores.

*Pg 3 Ln 83. The concentrations of CH4 standards used should be indicated, because although the average concentration is 51.2 _ 84.2nM, there were some sporadic samples with very high concentrations (more than 600 nM) and those concentrations should be within the calibration line.*

The measurements last for more than a decade and the standard gases we used changed several times. We have adjusted the concentrations of standard gases for every measurement to make sure that the values of the samples fall in the range of the calibration curves. In this case, we think it is not necessary to list all the $CH_4$ standards.

We will include this information in the method section.

*Pg 3Ln 89-90. The accurate in dissolved oxygen and Chla measurements should be indicated. How were temperature and salinity measured? What was the precision of these measurements?*

Temperature and salinity were measured by the sensors equipped on CTD. Different methods might be adopted for individual parameter during the past decades. A more comprehensive overview of temperature, salinity, dissolved oxygen, Chla as well as other parameters at BE can be found in the paper by Lennartz et al. (Lennartz, S. T., Lehmann, A., Herrford, J., Malien, F., Hansen, H. P., Biester, H., and Bange, H. W.: Long-term trends at the Boknis Eck time-series station (Baltic Sea), 1957–2013: does climate change counteract the decline in eutrophication? Biogeosciences, 11, 6323–6339, https://doi.org/10.5194/bg-11-6323-2014, 2014).

We will include this information in the method section.

*Pg 4 Ln 112-113. What H2S concentrations were measured? It would be interesting to include these values*

Unfortunately we did not measure the H$_2$S concentrations. The presence of H$_2$S was recognized by the strong smells of the bottom water.

We will add it to the text.

*Pg 4 Ln 115. Indicate the value of the DO concentration that was obtained in the surface waters. This upwelling has also been appreciated in other variables such as nutrients?*

We have shown the approximate O$_2$ saturation in the surface water, which is a better proof than the actual DO concentration. The occurrence of the upwelling can be identified by nutrients, too, but not as clearly as indicated by temperature and O$_2$. In this case, we think it is not necessary to include the variation of nutrients in the water column.

We will add more detail here.

*Pg 4 Ln 115-116. Since the authors write about behaviour of temperature and salinity, it would be convenient to include graphs of these variables in Figures 2 and 3.*

The variations of temperature and salinity will be shown in figure 2 and 3.

*Pg 5 Ln 124. What is the reason that in BE the Chla has elevated concentrations only in the upper layers in March and not occupied the whole water column as other works realised in this system? Are Chla and Secchi depth well related to the entire study?*

*Perhaps it could be included in the figure of the Chla the Secchi depth graph. If we look at figure 2 in 2006 and 2012 the Chla occupies the entire water column. What could have happened in these years for the Chla behaviour to be different?*

It remains unclear why elevated Chla concentrations were only detected in the upper layers in March. The overall correlation between Chla and Secchi depth is poor (r$^2$=0.17, p<0.0001, n=111). High Chla concentrations all over the water column in November/December 2006 and March 2012 were coinciding with slightly enhanced nutrients and high temperatures. Nutrients and temperature might be potential environmental controls on Chla distribution.

We will incorporate this additional information in section 4.1.

*Pg 5 Ln 130. To show seasonal and inter-annual variations, a table could also be presented showing the variation interval and annual mean value of each variable. Figures 2 and 3, although very illustrative, have been made with interpolations and do not show the specific data that it is interesting to know.*

A table will be shown in the main text as suggested.

*Pg 5 Ln 154. Is there any work in the area where CH4 benthic fluxes have been measured? If so, it would be interesting to include the value.*

There are several papers reporting benthic CH$_4$ fluxes in the Eckernförde Bay. Sedimentary CH$_4$ release, via pockmarks or ebullition, was discussed in section 4.3.

*Pg 6 Ln 166-167. Was the water more turbid? Did Secchi's disc reach less depth?*

We did not see a strong decline in Secchi depth.

We will add this information in the text.

*Pg 6 Ln 178. What is the r2 of the relation between salinity and CH4?*

r$^2$=0.84, and this value together with p and n are now included in the text.

*Pg6 Ln 184. Include variation of dissolved oxygen values to change from hypoxic to oxic condition in the bottom layer.*

The values are now included as suggested.

*Pg 6 Ln 189. Include the correlation coefficient (r2) of the relation between salinity and CH4 in November 2013. Pg 6 Ln 199. Include the correlation coefficient (r2) of the relation between salinity and CH4 in March 2014.*

r$^2$, p and n are now described in the text.

*Pg 7 Ln 213-214. Since CH4 saturation has been obtained from the surface methane concentration and equilibrium concentrations of CH4 in seawater, it is obvious that surface CH4 saturations are directly proportional to its concentrations in the surface water, I would omit this from the manuscript.*

The calculation of CH$_4$ saturation involves concentration, as well as temperature and salinity. The authors would like to point out that surface CH$_4$ saturations are directly proportional to its concentrations, despite of the significant seasonality in temperature and salinity. In another paper about dissolved N$_2$O at BE, the influence of temperature is stronger. We think this comparison might be interesting.

*Figures:*

*In the figures the letters and numbers are in Arial and not in Times New Roman like the rest of the manuscript.*

BG does not require a consistency for the fonts in the figures and in the text. As long as the fonts in all of the figures are consistent, it would not be a problem.

*Figure 1. The quality of the figure must be improved. Figures 2 and 3. The axis titles should appear with capital letters as: Depth, not depth and Dissolved oxygen not only oxygen. It should be convenient to include isolines in these figures for a better appreciation of the concentration variations. Figures with temperature and salinity variations should be included.*

The figures will be revised as suggested, except for the isoclines. We tried to include isoclines in figure 2 and 3, but they seem crowded and messy because of the strong gradients.

According to the comments from the reviewers, the following changes are made in the manuscript:

1. Include a table of inter-annual variations of temperature, salinity, wind speed and dissolved $CH_4$ concentrations.

2. Improve the quality of Fig. 1.

3. Show inter-annual and seasonal variations of temperature and salinity in Fig. 2 and 3, respectively.

4. Add more details about standard gases (in lines 85–86) and the overview of other parameters (in lines 94–95).

5. Include the information about $H_2S$ recognition (in line 117).

6. Add more details about upwelling signal from nutrients in lines 121–122.

7. Analyze the potential correlations between Chlorophyll *a* and nutrients or temperature in lines 130–133.

8. Compare $CH_4$ concentrations from the North Sea and the Eckernförde Bay in lines 171–175.

9. Briefly discuss the potential $CH_4$ contribution from porewater exchange and sediment resuspension in lines 180–183.

10. Reanalyze the impact of ebullition as a result of water level change and compare the values with our results in lines 188–196.

11. Add the specific numbers about correlations and $O_2$ concentrations in lines 200, 206, 212 and 219.

12. Few typos in the manuscript were corrected.

[revised manuscript text omitted]
 concentrations of standard gases were adjusted for every measurement to make sure that the values of the samples fall in the range of the calibration curves. The standard gas mixtures were calibrated against NOAA primary gas standard mixtures in the laboratory of the Max-Planck-Institute for Biogeochemistry in Jena, Germany. Further details about the measurements and calculations of the dissolved $CH_4$ concentration can be found in Bange et al. (2010). The mean precision of the $CH_4$ measurements, calculated as the median of the estimated standard errors (see David, 1951) from all triplicate measurements, was ± 1.3 nM. Samples with an estimated standard error of >10 % were omitted. Dissolved $O_2$ concentrations were measured with Winkler titrations, and Chlorophyll *a* concentrations were measured with a Fluorometer (Grasshoff et al., 1999). Secchi depth was measured with a white disk

(~30 cm in diameter). Sea levels were measured at Kiel-Holtenau, which is about 15 km away from the BE time-series station (http://www.boos.org/). A more comprehensive overview of temperature, salinity, dissolved $O_2$, Chlorophyll *a* as well as other parameters at the BE time-series station can be found in Lennartz et al. (2014).

**3.2 Calculation of saturation and air-sea flux density**

The CH$_4$ saturation (S$_{CH4}$, %) was calculated as:

$$S_{CH4} = 100 \times CH_{4obs}/CH_{4eq} \tag{1}$$

[revised manuscript text omitted]

**4.2 Enhanced $CH_4$ concentrations in the upper water layer**

In agreement with Schmale et al. (2010) and Bange et al. (2010), we found that $CH_4$ concentrations generally increase with water depth, indicating a prevailing release of $CH_4$ from the sediments into the water column in the Baltic Sea (see Sect. 4.1). Nonetheless, unusual high $CH_4$ concentrations in the upper layers were detected sporadically at the BE time-series station during 2006–2017 (Fig. 2). In November 2013 and March 2014, average $CH_4$ concentrations in the upper waters were 187.2 ± 13.9 nM (1–10 m) and 217.8 ± 1.4 nM (5–10 m), which are about 16 and 5 times higher than those found in the bottom layers, respectively (Fig. 4). The most striking event occurred in December 2014, when $CH_4$ concentrations in the upper layer (1–15 m) were as high as 692.6 ± 3.4 nM (19,890 ± 115 %), whereas dissolved $CH_4$ in the bottom layer (20–25 m) was ~50 nM. The surface $CH_4$ concentration in December 2014 was the highest observed during 2006–2017. In December 2014, a major Baltic inflow (MBI) event occurred, carrying large amounts of saline and oxygenated water from the North Sea into the Baltic Sea (Mohrholz et al., 2015). Dissolved $CH_4$ concentrations in the surface North Sea were much lower than in the Eckernförde Bay (Bange et al., 1994; Rehder et al., 1998), and therefore a direct $CH_4$ contribution from the North Sea by oxygenated waters seems unlikely. We hypothesize that this inflow substituted the lower part of the water column which had high $CH_4$ concentration throughout the water depth before, opposite to, e.g., an in-situ production of $CH_4$ at the surface being responsible for the observed concentration profile anomaly. The MBI is the third strongest event ever recorded, and an unusual outflow period was detected in the Eckernförde Bay: Sea levels declined since mid-November and reached minimum on 10 December, and then began to increase with the inflow (Fig. 5). The sampling at the BE time-series station took place on 16 December, during the main inflow period. Extreme weather conditions (wind speed >15 m s$^{-1}$) were observed several days before the sampling date, and storm-generated waves and currents could have affected the sediment structures in the Eckernförde Bay (Oris et al., 1996). Currents across the seabed can result in pressure gradients that drive porewater flow within the permeable sediments (Ahmerkamp et al., 2015), which might be a potential $CH_4$ source. Sediment resuspension might also contribute to enhanced $CH_4$ release, but we did not observe a significant decline in Secchi depths in December 2014 (Fig. 2).

The significant decrease in sea level alleviated the static pressure on the sediments. Enhanced $CH_4$ release from the sediments, via gas bubbles or exchange from porewater, may have led to the accumulation of $CH_4$ in the water column. Similar hydrostatic pressure effects were also reported in tidal systems such as mangrove creeks and estuaries (see e.g. Barnes et al. 2006; Maher et al., 2015; Sturm et al., 2017). Atmospheric pressure also contributes to the overall pressure on the sediments, but it is not recorded at the BE time-series station and thus was omitted. Although the water level fluctuation

of ± 1 m (Fig. 5) seems rather small compared to the water depth (28m), it might exert a strong influence on the sediments. Water level fluctuation, when there was no strong wind or inflow event, was approximately ± 0.2 m in the Eckernförde Bay. Lohrberg et al. (2020) detected a change in water level (± 0.5 m) and air pressure (± 1500 Pa, equivalent to approximately ± 0.15 m of water level fluctuation) during a weak storm in the fall of 2014. The fluctuation in hydrostatic pressure induced a pronounced $CH_4$ ebullition event in the Eckernförde Bay, and a sedimentary $CH_4$ flux of 1916 µmol m$^{-2}$ d$^{-1}$ was estimated (Lohrberg et al., 2020).  This value is generally in good agreement with the sharp increase in the sea-to-air $CH_4$ fluxes in December 2014 (see section 4.3). 
[revised manuscript text omitted]

505

Table 1. Annual mean (arithmetic average ±standard deviation) of water temperature, salinity, wind speed and dissolved $CH_4$ concentrations at the BE time-series station during 2006–2017. Water temperatures, salinity and $CH_4$ concentrations were averaged over the water column (0–25 m). Wind speeds were recorded at Kiel lighthouse.

| year | Temperature (°C) | Salinity | Wind speed ($u_{10}$, m s$^{-1}$) | $CH_4$ (nM) |
|---|---|---|---|---|
| 2006 | 9.19±5.75 | 20.14±3.11 | 7.5±2.6 | 39.3±38.1 |
| 2007 | 9.68±4.55 | 17.78±2.14 | 7.5±2.5 | 44.9±45.5 |
| 2008 | 10.11±4.20 | 19.14±3.43 | 6.2±2.1 | 36.9±41.9 |
| 2009 | 9.20±4.81 | 18.36±2.22 | 7.3±2.3 | 27.8±26.2 |
| 2010 | 8.47±5.20 | 17.80±3.22 | 5.5±2.7 | 34.8±39.3 |
| 2011 | 8.74±5.16 | 19.14±2.78 | 6.8±3.1 | 36.9±29.1 |
| 2012 | 9.47±3.89 | 18.67±2.63 | 8.7±2.1 | 46.4±44.3 |
| 2013 | 9.04±5.45 | 17.89±3.74 | 5.9±2.8 | 67.7±83.1 |
| 2014 | 10.38±4.93 | 19.17±2.79 | 7.0±3.3 | 101.4±183.3 |
| 2015 | 9.19±4.28 | 19.71±3.30 | 6.1±2.8 | 35.7±36.3 |
| 2016 | 10.09±4.71 | 18.80±3.19 | 5.9±1.7 | 52.6±111.4 |
| 2017 | 10.21±4.86 | 19.50±2.11 | 6.8±2.4 | 30.5±22.9 |

[Figure]

**Fig. 1 Location (black square) of the Boknis Eck time-series station in the Eckernförde Bay, southwestern Baltic Sea. (from Hansen et al., 1999)**

510

[Figure]

[Figure]

**Fig 2. Distributions of temperature, salinity, dissolved O$_2$, Chlorophyll *a* and CH$_4$ at the BE time-series station during 2006–2017. Black dots indicate the monthly measurements of Secchi depth. To get a better visualization, the maximum color bar for CH$_4$ concentration is 300 nM, but some of the actual concentrations are higher (for example, in December 2014 and in autumn 2016).**

515

[Figure]

[Figure]

**Fig 3. Mean seasonal variations of temperature, salinity, dissolved O₂, Chlorophyll *a* and CH₄ at the BE time-series station during 2006–2017. CH₄ concentrations in December 2014 were excluded in plotting.**

[Figure]

520

**Fig 4. Vertical distribution of Chlorophyll *a*, salinity and CH₄ concentrations in the water column in December 2014 (a), November 2013 (b) and March 2014 (c).**

[Figure]

525

**Fig 5. Sea level variations in November and December, 2014. The black line indicates the occurrence of BE sampling in December 2014.**

[Figure]

**Fig 6. Inter-annual variations of dissolved CH$_4$ concentration (a), saturation (b) and flux density (c) at the BE time-series station during 2006–2017. Data collected from December 2014 were not shown.**

[Figure]

**Fig 7. Comparison of surface CH$_4$ saturations (a) and flux densities (b) from time-series stations of BE, Saanich Inlet (SI) and ALOHA. For the computation of flux density, the equations of Nightingale et al. (2000) and Wanninkhof (2014) were used for SI and ALOHA, respectively. Data in December 2014 at the BE time-series station were not included. Please note the break on the y axis for both charts.**